# Multiple Pulmonary Involvement in the Rapidly Progressive Evolution of Rheumatoid Arthritis

**DOI:** 10.3390/diagnostics14192175

**Published:** 2024-09-29

**Authors:** Beatrice Mahler, Mădălina Ioana Moșteanu, Raluca Bobocea, Iris Negoescu, Leonard Florentin Mircea, Adrian Tudor, Maria Teodora Bogdan, Alina Croitoru, Angela Stefania Marghescu

**Affiliations:** 1Cardiothoracic Department, Faculty of Medicine, “Carol Davila” University of Medicine and Pharmacy, Bucharest, 020956 Bucharest, Romania; beatrice.mahler@umfcd.ro (B.M.); maria.teodora.bogdan@gmail.com (M.T.B.); 2Pneumology Department, “Marius Nasta” Institute of Pneumology, 050159 Bucharest, Romania; madalina.mosteanu@yahoo.com (M.I.M.); raluca.bobocea@marius-nasta.ro (R.B.); iris.negoescu@gmail.com (I.N.); 3Department of Diabetes, Nutrition and Metabolic Diseases, University of Medicine and Pharmacy of Craiova, 200349 Craiova, Romania; 4Pathology Department, “Marius Nasta” Institute of Pneumology, 050159 Bucharest, Romania; adrian.tudor@marius-nasta.ro (A.T.); angela.varban@drd.umfcd.ro (A.S.M.)

**Keywords:** rheumatoid arthritis, secondary interstitial pneumonia, bronchiectasis, pulmonary embolism

## Abstract

A 66-year-old man, a 40-year smoker, was diagnosed with rheumatoid arthritis in 2018. He was treated for one year with methotrexate, and, later in 2020, he was diagnosed with interstitial pulmonary fibrosis. In 2022, treatment with nintedanib was initiated, with clinical improvement being indicated but without showing a functional or imaging benefit. The evolution of the disease was rapidly progressive and unfavorable, with death occurring due to pulmonary thromboembolism. Following the autopsy, triple lesions of the RA at the lung level were confirmed: interstitial, of the NSIP type with a fibrosing character at the level of the lower airways of the bilateral bronchiectasis type, and vascular damage due to pulmonary thromboembolism secondary to chronic inflammation.

Rheumatoid arthritis (RA) is a systemic autoimmune disease that manifests as a chronic inflammatory condition, affecting 0.5–1% of the population. It is three times less common in men [1] and causes extra-articular manifestations in about 50% of patients [2]. The inflammatory damage is widespread, directly impacting the respiratory function through damage to the pleura, lung parenchyma, airways, lymph nodes, and blood vessels, and also indirectly occurring through damage to the bone skeleton and rib cage muscles, leading to various respiratory symptoms [3]. Multiple risk factors contribute to the development of RA-associated interstitial lung disease (RA-ILD), with the most consistent associations found in studies being older age, male sex, a history of smoking, rheumatoid factor (RF) seropositivity, and anti-cyclic citrullinated peptide antibodies (anti-CCP) [4]. Smoking is a significant cofactor for lung damage in RA [5], and it is the primary risk factor (found in over 90% of cases) for desquamative interstitial pneumonia (DIP). RA-ILD significantly increases the mortality risk from 3.8% in patients with RA to 13.9% in those with RA-ILD [6,7].

Literature findings state that the main histopathological lesions present include non-specific interstitial pneumonia (NSIP) in 30% of cases and the usual interstitial pneumonia in 40% of cases (UIP) [8]. Park et al. al suggested that patients with UIP patterns have a lower life expectancy compared to patients with NSIP-type patterns [9].

We report the case of a 66-year-old man, a former smoker with a history of high blood pressure and basal cell carcinoma, who was first diagnosed in 1997 and then, subsequently, treated surgically in 2007 and 2018. In 2018, he was diagnosed with seropositive anti-CCP positive rheumatoid arthritis (RA) at a stage I anatomopathological classification and Steinbrocker functional class I. Treatment with methotrexate was initiated at this time, starting with a dose of 10 mg administered subcutaneously per week, which was progressively increased to 20 mg per week. After three years, in 2021, he was diagnosed with interstitial lung disease (ILD) and chronic hypoxemic respiratory failure, prompting a change in his RA treatment regimen to include Plaquenil, sulfasalazine, medrol (4 mg), and oxygen therapy (Figure 1 and Figure 2). In 2022, treatment with nintedanib was started, leading to a favorable clinical response, although there were no functional or imaging improvements noted.

The patient passed away following a decompensatory episode in a viral context that necessitated hospitalization; bacteriological sputum tests were negative (Figure 3 and Figure 4). During his hospital stay, he experienced severe hypoxemia despite non-invasive ventilation with continuous positive airway pressure (CPAP), maintaining a SpO2 level of 80%. He was hemodynamically stable and afebrile. On the third day of hospitalization, his oxygen saturation suddenly dropped; polypnea and tachypnea were among his clinical manifestations, thus necessitating intubation and mechanical ventilation. Subsequently, he developed progressive bradycardia that was unresponsive to treatment, and he ultimately succumbed to unresponsive cardio-respiratory arrest.

A necropsy was performed, revealing pulmonary thromboembolism as the cause for the patient’s sudden deterioration. A gross examination of the lung showed bronchiectasis, peribronchiectatic pulmonary condensations, extensive fibrotic changes, and cystic spaces (which was more evident in the subpleural zone). Additional findings included complicated aortic atheromatosis, a simple serous cyst on the left kidney, and plurivisceral stasis.

The pathology report was signed as “pulmonary thromboembolism and interstitial lung disease (UIP pattern and NSIP pattern)” in the context of a patient with rheumatoid arthritis undergoing treatment. The microscopic lesions exhibited extensive fibrosis and moderate lymphocytic inflammatory infiltrate within the interstitium, traction bronchiectasis (NSIP pattern), and peribronchiectatic pneumonia, along with enlarged respiratory spaces, some of which were bordered by a significant fibrotic tissue (‘honeycombing’) UIP pattern.

The patient’s smoking history could also contribute to lung alterations with smoking being associated with respiratory bronchiolitis, desquamative interstitial pneumonia-like (DIP-like) changes, smoking-related interstitial fibrosis (SRIF), emphysema, and even vascular remodeling. SRIF is a distinct form of hyalinized collagen deposition in the interstitial compartment, predominantly affecting the subpleural and centrilobular regions, with minimal clinical symptoms. This differs from our patient’s case, which involved extensive fibrosis with significant clinical manifestations.

Regarding rheumatoid arthritis (RA), it is indeed recognized that smoking is a major environmental risk factor for RA development and can exacerbate its progression. The multifactorial nature of the patient’s condition made it very difficult to determine the extent to which each factor could have contributed to these lung alterations.

Even if ILD due to methotrexate therapy is rare and is mostly described as NSIP, aspects of fibrotic hypersensitivity pneumonitis-like lesions have been observed, which are exacerbated by the active inflammatory lung disease associated with rheumatoid arthritis (RA) [10,11,12]. The imaging features—ground glass opacities typical of nonspecific interstitial pneumonia (NSIP) combined with macroscopically visible subpleural cystic lesions—suggested a rapid progression to fibrosis and indicated an intense inflammatory process that is likely exacerbated by mechanisms related to RA and the inflammation induced by methotrexate [13].

In conclusion, the management of RA should carefully consider patient-specific risk factors when prescribing treatment, particularly concerning potential adverse effects on other organs directly or indirectly affected by RA-related inflammation. The association of the two aspects of UIP and NSIP imposes an open approach on the medical world, which also allows for the diagnosis of overlap syndromes in interstitial pathology. Pulmonary inflammation directly involves pulmonary vessels, initiating coagulation processes that can lead to fatal embolism. Although the clinical response to nintedanib was favorable, the patient ultimately succumbed to pulmonary embolism. We propose that international treatment protocols for such cases should include prophylaxis for pulmonary embolism to mitigate this risk (Figure 5, Figure 6, Figure 7, Figure 8, Figure 9, Figure 10 and Figure 11).

The presented case highlights an important discussion regarding the recommendation of methotrexate for a male smoker with rheumatoid arthritis (RA), who was positive for both rheumatoid factor (RF) and anti-CCP antibodies. This scenario presented a significant risk for the development of interstitial lung disease (ILD) and a high mortality rate. As observed, the patient developed ILD after three years of disease progression. The decision to initiate treatment with nintedanib was made following the appearance of usual interstitial pneumonia (UIP)-type lesions, and it is in line with the recommendations found in specialized literature for selected cases. The national guidelines also support starting treatment regardless of the DLco value. The total treatment duration was eight months, during which the patient’s clinical condition showed significant improvement.

A postmortem microscopic examination of the lung lesions revealed fibrotic lung disease, with signs of pulmonary hypertension (that were characterized by medial hypertrophy of the pulmonary arteries) and inflammatory infiltrates (which were predominantly lymphocytic with a peribronchial distribution). Some histopathological aspects may be similar to hypersensitivity pneumonitis with fibrotic evolution, which may give rise to the hypothesis that methotrexate therapy may have had a toxic effect on this patient. Combined with the active inflammatory lung disease associated with RA, this likely contributed to a pattern of rapid fibrosis and, ultimately, death.

While certain studies have proposed an association between cumulative methotrexate dosage and the development of interstitial lung disease in the context of rheumatoid arthritis (RA), the relationship has not been definitively established. Further studies are needed to clarify this aspect. However, it is crucial to acknowledge the challenge of differentiating between drug-induced and disease-related pulmonary manifestations. A recent meta-analysis emphasized disease activity as a primary cause of ILD, and it noted potential biases in attributing these changes solely to methotrexate. However, other meta-analyses that specify the use of methotrexate may not be related to association but to an acute exacerbation in RA with ILD [14]. This underscores the importance of considering both the disease’s natural course and potential drug effects when interpreting histopathological findings.

The presence of vascular damage in an acute inflammatory context, which was also highlighted in the pathological findings along with chronic pulmonary hypertension, created conditions that were conducive to the development of massive pulmonary thromboembolism despite the correct initiation of anticoagulant therapy at the time of admission.

## Figures and Tables

**Figure 1 diagnostics-14-02175-f001:**
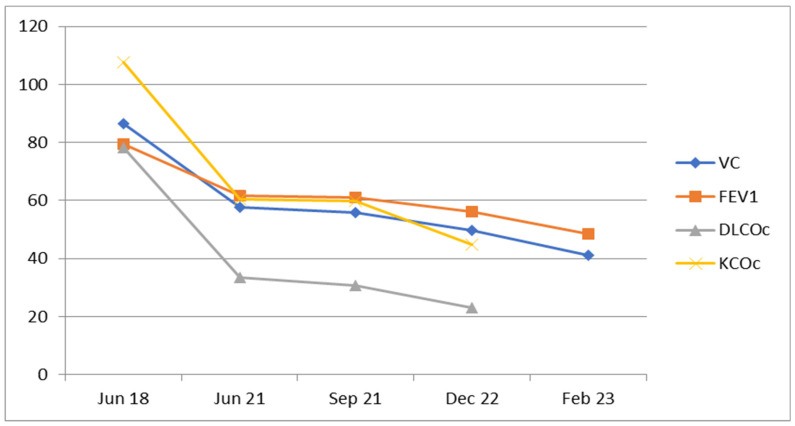
A graph of a progressive decline in lung function, vital capacity (VC), diffusing capacity of the lungs for carbon monoxide (DLco), and carbon monoxide transfer coefficient (Kco), with a rapid drop shown in the first 3 years after diagnosis, which is then followed by a much slower rate of decline.

**Figure 2 diagnostics-14-02175-f002:**
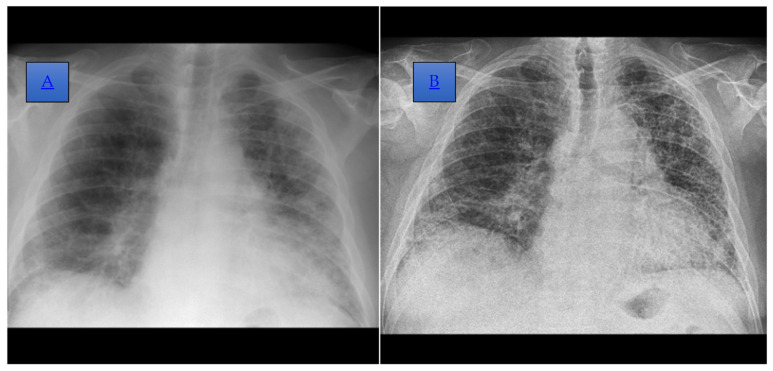
(**A**) Posteroanterior incidence chest radiograph, performed in 2021, which highlights the accentuation of the bilateral juxtahilar and basal, interstitial, and peribronchovascular pattern, which is more obvious on the left side. A reduction in lung transparency bilaterally, more obviously the left subpleural, is also shown. (**B**) Posteroanterior incidence chest radiograph, performed in 2022, where an accentuation of the perilobular interstitium with a reticular and pseudo-honeycomb appearance and on the basal right thickening of the pleuritic septa, dispersed in the left 2/3 lobe and the external basal in the right lobe, is shown.

**Figure 3 diagnostics-14-02175-f003:**
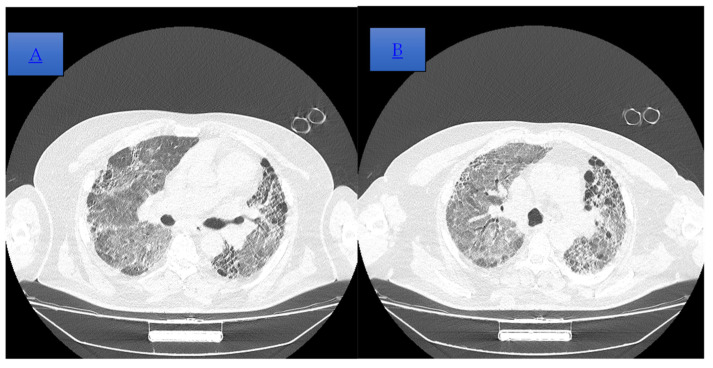
(**A**,**B**) Cranio-caudal dispersed reticulonodular opacities, which associate the traction and cylindrical bronchiectasis, on the right lung and cystic on the left basal lung, with an imaging aspect that illustrates end-stage pulmonary fibrosis (UIP pattern). The association of extensive ground-glass opacities is associated with the extensive inflammatory process (NSIP pattern). Global cardiomegaly with a dilated pulmonary trunk shown.

**Figure 4 diagnostics-14-02175-f004:**
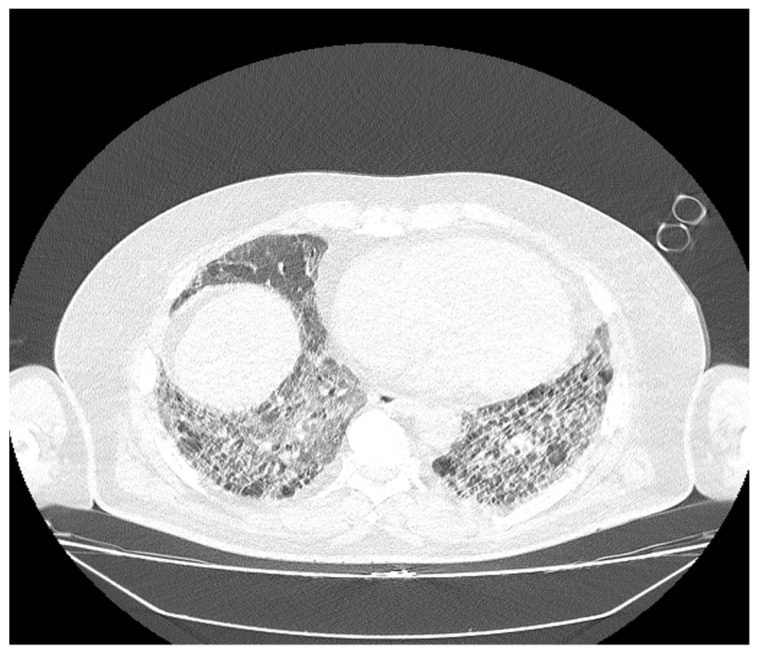
The bilateral pulmonary bases presented dispersed reticulonodular opacities, which associate with traction bronchiectasis: an imaging aspect that illustrates end-stage pulmonary fibrosis (UIP pattern). The association of extensive ground-glass opacities is associated with the extensive inflammatory process (NSIP pattern).

**Figure 5 diagnostics-14-02175-f005:**
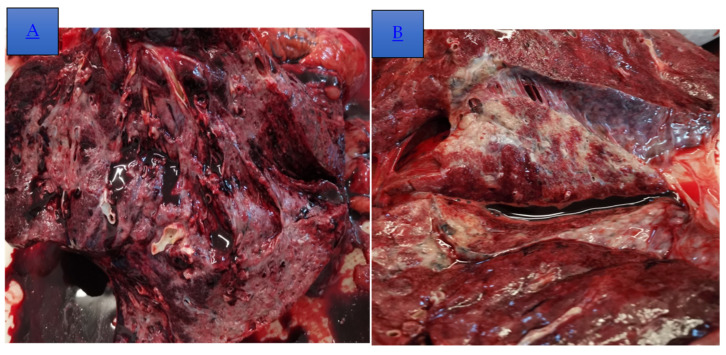
(**A**) Lung parenchyma had the decreased elasticity and tactile fremitus, with diffuse areas of condensation that were brown-grey colored; on the cut section, an important quantity of dark red blood (a hallmark of hyperemia) can be observed. (**B**) The lung parenchyma had modified the gross aspect, with multiple, poorly circumscribed, grey areas of increased consistency, hyperemia, and adherences between different visceral pleura zones; the elasticity of the lung parenchyma and tactile fremitus were diminished.

**Figure 6 diagnostics-14-02175-f006:**
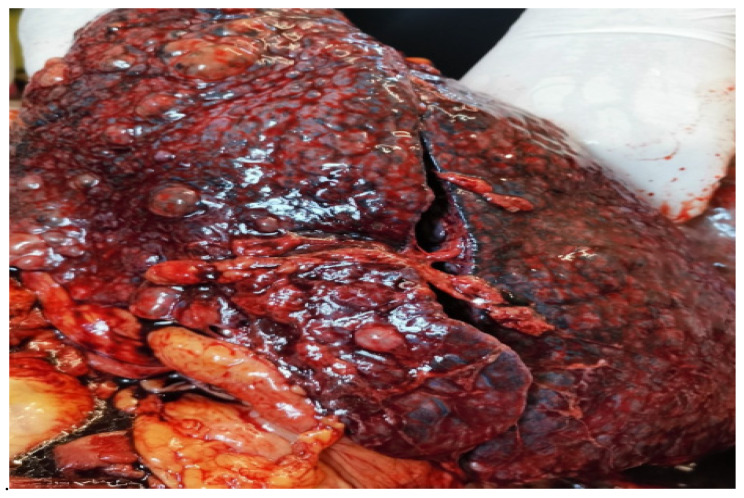
The lung parenchyma had multiple cystic spaces with aeric content, and it was localized throughout the subpleural zone.

**Figure 7 diagnostics-14-02175-f007:**
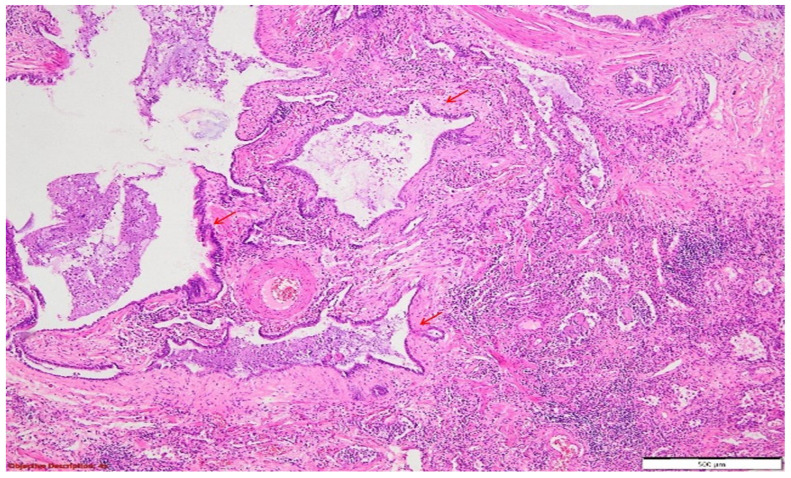
Pulmonary tissue with an enlarged bronchiolar lumen (bronchiectasis →), containing mucus and rare inflammatory elements; smooth muscle hyperplasia of the arterial media, and important polymorphous inflammatory infiltrate (predominantly lymphocytic) in the peri-bronchiolectatic interstitial parenchyma. HE, 40×.

**Figure 8 diagnostics-14-02175-f008:**
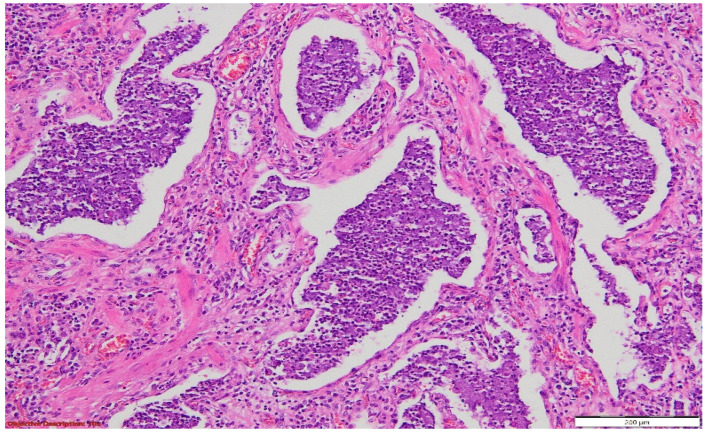
Lung parenchyma with distended alveoli by abundant inflammatory elements composed mainly of neutrophils; thickened interstitial space due to the hyperemia and moderate inflammation. HE, 100×.

**Figure 9 diagnostics-14-02175-f009:**
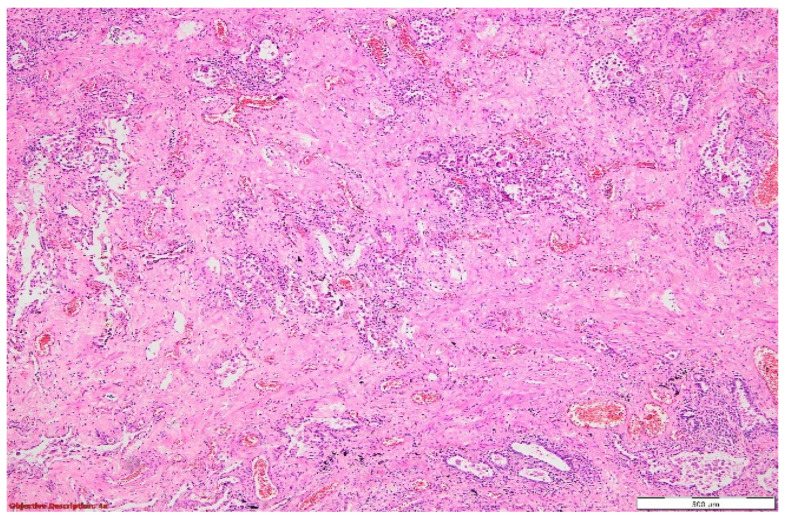
Lung tissue with architectural distortion, enlarged interstitial compartment due to significant fibrosis, dilated capillaries (with a high number of red blood cells in the lumen (hyperemia)), and mild inflammatory cells (mostly lymphocytes). The hyperplasia of the pneumocytes was evident. HE, 40×.

**Figure 10 diagnostics-14-02175-f010:**
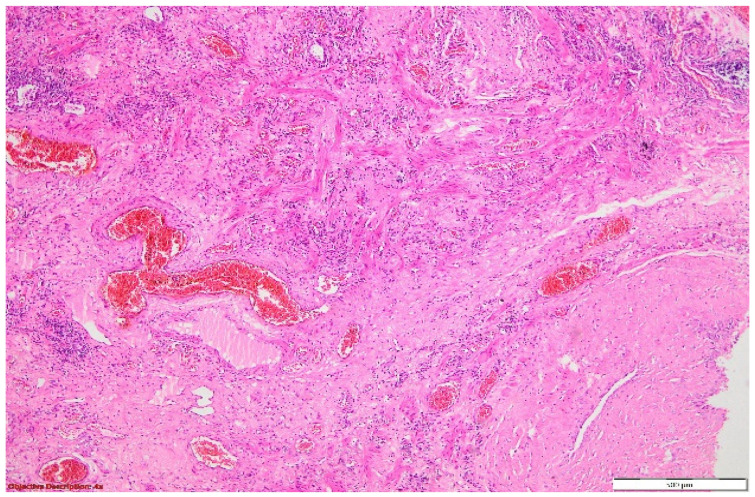
Significant smooth muscle proliferation in the pulmonary parenchyma was visible in the blood vessel’s tunica media and interstitial space. HE, 40×.

**Figure 11 diagnostics-14-02175-f011:**
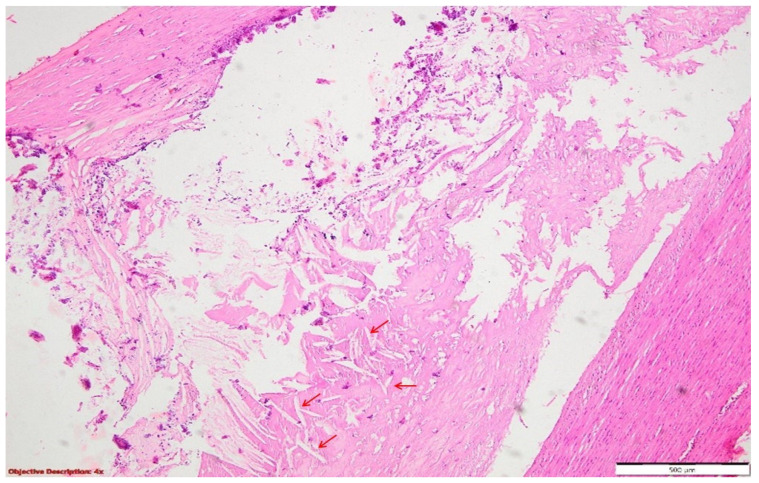
The atherosclerosis lesions in the thoracic aorta were characterized by a thick, fibro-sclerotic arterial wall, with an abundance of extracellular cholesterol deposits (→). HE, 40×.

## Data Availability

Not applicable.

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
