# Peer review of "Multiple Pulmonary Involvement in the Rapidly Progressive Evolution of Rheumatoid Arthritis"

_diagnostics, 2024, doi:10.3390/diagnostics14192175_

Round 1

Reviewer 1 Report

Comments and Suggestions for Authors

In this manuscript the Authors report a case of RA-ILD rapidly progressive. In general, it is not clear what this report is focused on. Consider to centre the report on a comparison between radiology and 

Comments

I strongly suggest an English native speaking revision. Moreover the case report must be organized in at least two sections: case presentation and comment/discussion. In the main text the case presentation is interposed between the comments.

Please, report the autoimmune profile and MTX posology.

The NSIP pattern is quite uncommon in RA-ILD and often related to good prognosis. Discuss this issue focusing on what had a detrimental impact in this patients.

PAR: explain the acronym

L15 “NSIP type with a fibrosing character”: what does it mean?

Fig. 1 What is VC? Provide FVC value?

In order to make the case report interesting for the reader I strongly suggest to show the radiological corresponding to the histological images

Comments on the Quality of English Language

I strongly suggest an English native speaking revision.

Author Response

We truly appreciate your effort in reviewing the case we submitted. Your comments were insightful and have significantly helped us improve the quality of the material for the publisher. We've corrected the English and acronyms used. Additionally, we specified the type of RA identified at diagnosis and the dose of methotrexate. We also addressed the clarifications related to the histopathologic findings in correlation with the imaging and clinical data. Furthermore, we added new CT images to support the reported data.

Reviewer 2 Report

Comments and Suggestions for Authors

Dear Editor,

The authors aim to present Multiple Pulmonary Involvement in the Rapidly Progressive Evolution of Rheumatoid Arthritis. Although it contains valuable images in terms of educational material, it does not contribute recent data to the literature.

Best Regards

Comments on the Quality of English Language

Minor editing of English language required.

Author Response

We appreciate your efforts in reviewing our proposed case. Your comments were relevant and have helped us improve the quality of the material submitted to the publisher. We consider the case as an important demonstration of the impact that rheumatoid arthritis has on the lungs, highlighting the inflammation it induces in the interstitium and small vessels. Additionally, it emphasizes the fatal role that coagulation activation, within the inflammatory context, can play through pulmonary thromboembolism.

Reviewer 3 Report

Comments and Suggestions for Authors

The review of the case report: Multiple pulmonary involvement in the rapidly progressive evolution of rheumatoid arthritis.

The authors describe patient diagnosed with  rheumatoid arthritis (RA) presenting with rapidly progressive lung fibrosis and  pulmonary embolism, with fatal outcome. The paper was submitted to the section: Interesting Images.

In my opinion, the paper deserves major revision.

My critical remarks concerning the images are as follows:

1.      The presented chest CT scans are not sufficient to recognise any type of fibrotic ILD, as only the upper and middle parts of the lungs were visualised.

Based on the presented figures, end stage lung fibrosis may be diagnosed, with honeycombing and bronchiectasis, and possibly with some ground glass opacities, that were not mentioned by the authors.   I suggest inclusion of chest CT reconstructions presenting  the whole lungs, in  frontal and sagittal projections, to show the distribution and character of lung fibrosis as well as the other signs of interstitial lung disease.

2.      As pulmonary embolism was diagnosed before patient’s death, pulmonary emboli should be also presented  on chest CT angiography.

3.      It is not necessary to include   three figures showing macroscopic view of the lungs taken during autopsy.

4.      Microscopic lung views show fibrotic lung disease, with signs of pulmonary hypertension ( pulmonary arteries medial hypertrophy) and inflammatory infiltrate (predominantly lymphocytic) with peribronchial distribution. Such pathologic view could be characteristic of the fibrotic type of hypersensitivity pneumonitis, nevertheless it may also be combined with  active inflammatory lung disease in the course of RA. Neither pulmonary emboli, nor lung infarcts were demonstrated on microscopic examination,  this needs explanation.

Critical remarks to the data presented by the authors:

1.      As far as I understand,  nintedanib was introduced in 2022. At that time, the patient presented with  end stage fibrotic lung disease, with DLCO of 23%, therefore according to current regulations, such patient should not receive anti-fibrotic therapy. What was the treatment duration?

2.      Lung fibrosis is classified as: reticular opacities +/- bronchiectasis +/-  honeycombing. They are not distinct types of lung fibrosis, but rather an increasing stages of fibrotic lung disease. Chest CT in NSIP shows lung fibrosis localized in lower parts of the lungs, with a frequent sparing of sub-pleural region and also, with coexisting ground glass opacities. Honeycombing is not found in NSIP, it is the characteristic feature of  UIP. Chest CT in UIP patients should not include excessive ground glass opacities, but  they may be present at the times of acute disease worsening.  RA pneumonitis may present as UIP, NSIP, OP. The authors should indicate the radiologic and pathologic features of NSIP or UIP in the presented case report.

3.      The author mentioned that pulmonary embolism was diagnosed before death. Was it treated and how?

4.      The figure representing respiratory indices: please use VC instead of CV

Author Response

We appreciate your effort in reviewing our proposed case. Your comments were pertinent and have helped us improve the quality of the material.

  1. We've added CT images of the lower lobes.
  2. The diagnosis of PTE was established postmortem, and we've included these details in the case presentation.
  1. We've reduced the number of macroscopic images, but we believe they offer valuable insights for readers, particularly because these postmortem findings are quite rare.
  2. We've added the explanations you suggested.
  3. National guidelines allow the use of nintedanib, and the patient also received a second opinion from another EU country that recommended the same treatment. The positive clinical outcome throughout the treatment supports the use of nintedanib. Since the patient’s death was due to pulmonary thromboembolism (PTE), I believe international protocols should include PTE prophylaxis in these cases.
  4. We've corrected the English and acronyms used.

Round 2

Reviewer 1 Report

Comments and Suggestions for Authors

- Please provide literature supporting this statement

"The histopathological appearance was suggestive of hypersensitivity pneumonitis with fibrotic evolution, which supports the hypothesis that methotrexate therapy may have had a toxic effect on this patient."

The recent meta-analysis show that the disease activity is the actual cause of ILD and MTX is often a bias.

- "The presence of vascular damage in an acute inflammatory context, also highlighted in the pathological findings, along with chronic pulmonary hypertension, created conditions that were conducive to the development of massive pulmonary thromboembolism"

The patients was a lung neoplastic smoker: the lung alteration can easily be related to this condition. Even RA seems to be related to the smoke habit. In order to understand if a short-standing RA could have contribute to the lung alteration it is mandatory to know the evolution of disease activity.

Author Response

- Please provide literature supporting this statement

"The histopathological appearance was suggestive of hypersensitivity pneumonitis with fibrotic evolution, which supports the hypothesis that methotrexate therapy may have had a toxic effect on this patient."

The recent meta-analysis show that the disease activity is the actual cause of ILD and MTX is often a bias.

Thank you for your valuable comment!

While disease activity can indeed contribute to interstitial lung disease (ILD), there is evidence suggesting methotrexate-induced lung toxicity as a recognized entity. Methotrexate-induced lung toxicity, including hypersensitivity pneumonitis, has been documented in multiple studies. For instance:

- Fujimori T, Sano C, Ohta R. Drug-Induced Hypersensitivity Syndrome Due to Long-Term Usage of Methotrexate: A Case Report. Cureus. 2024 Apr 20;16(4):e58659. doi: 10.7759/cureus.58659. PMID: 38770448; PMCID: PMC11105963.

‘Methotrexate (MTX), a cornerstone treatment for rheumatoid arthritis (RA), is associated with drug-induced hypersensitivity syndrome (DIHS), including rare instances of methotrexate-induced pneumonitis’

‘Methotrexate (MTX) is known as a drug that causes drug-induced hypersensitivity syndrome (DIHS). One expression of DIHS is drug-induced pneumonitis.’

‘Pneumonia caused by MTX is called methotrexate-induced pneumonitis. Although this mechanism has yet to be completely elucidated, several hypotheses have been proposed. Methotrexate alters the immune system, including interleukin-8, which can induce autoimmune and hypersensitivity reactions and lead to pneumonia (…).’

‘Although most MTX-related DIHS is said to occur within a few months, it is uncommon to develop it after long-term administration, such as years and decades.’

- Juge PA, Lee JS, Lau J, Kawano-Dourado L, Rojas Serrano J, Sebastiani M, Koduri G, Matteson E, Bonfiglioli K, Sawamura M, Kairalla R, Cavagna L, Bozzalla Cassione E, Manfredi A, Mejia M, Rodríguez-Henriquez P, González-Pérez MI, Falfán-Valencia R, Buendia-Roldán I, Pérez-Rubio G, Ebstein E, Gazal S, Borie R, Ottaviani S, Kannengiesser C, Wallaert B, Uzunhan Y, Nunes H, Valeyre D, Saidenberg-Kermanac'h N, Boissier MC, Wemeau-Stervinou L, Flipo RM, Marchand-Adam S, Richette P, Allanore Y, Dromer C, Truchetet ME, Richez C, Schaeverbeke T, Lioté H, Thabut G, Deane KD, Solomon JJ, Doyle T, Ryu JH, Rosas I, Holers VM, Boileau C, Debray MP, Porcher R, Schwartz DA, Vassallo R, Crestani B, Dieudé P. Methotrexate and rheumatoid arthritis associated interstitial lung disease. Eur Respir J. 2021 Feb 11;57(2):2000337. doi: 10.1183/13993003.00337-2020. PMID: 32646919; PMCID: PMC8212188.

‘However, other than acute or subacute hypersensitivity pneumonitis, which is a rare complication of MTX, the evidence for a cause-and-effect relationship in modern populations between MTX and chronic fibrotic ILD in a patient with RA (i.e. RA-ILD) is unsettled. ’

- Kiely P, Busby AD, Nikiphorou E, Sullivan K, Walsh DA, Creamer P, Dixey J, Young A. Is incident rheumatoid arthritis interstitial lung disease associated with methotrexate treatment? Results from a multivariate analysis in the ERAS and ERAN inception cohorts. BMJ Open. 2019 May 5;9(5):e028466. doi: 10.1136/bmjopen-2018-028466. PMID: 31061059; PMCID: PMC6501950.

‘A hypersensitivity pneumonitis is a rare adverse effect of MTX described in 0.43% (…) This usually occurs early, within the first year of treatment, but has been reported up to 3 years after starting MTX.’

- Dana Oprea A. Chemotherapy Agents With Known Pulmonary Side Effects and Their Anesthetic and Critical Care Implications. J Cardiothorac Vasc Anesth. 2017 Dec;31(6):2227-2235. doi: 10.1053/j.jvca.2015.06.019. Epub 2015 Jun 12. PMID: 26619953.

‘Two to eight percent of patients receiving methotrexate develop evidence of lung toxicity. The toxicity usually is evident shortly after intravenous or intrathecal use, but may not appear until after months of oral use. In most cases, symptoms occur within a year of methotrexate initiation. The most common presentation is hypersensitivity pneumonitis’

While certain studies have proposed an association between cumulative methotrexate dosage and the development of interstitial lung disease in the context of rheumatoid arthritis (RA), this relationship is not definitively established. Further studies are needed to clarify this aspect.

However, it is crucial to acknowledge the challenge of differentiating between drug-induced and disease-related pulmonary manifestations. A recent meta-analysis, as you mentioned, emphasizes disease activity as a primary cause of ILD and notes potential biases in attributing these changes solely to methotrexate. This underscores the importance of considering both the disease's natural course and potential drug effects when interpreting histopathological findings.

We adjusted the manuscript to reflect these nuances and include references supporting both perspectives, ensuring a balanced view on the role of methotrexate in this context.

- "The presence of vascular damage in an acute inflammatory context, also highlighted in the pathological findings, along with chronic pulmonary hypertension, created conditions that were conducive to the development of massive pulmonary thromboembolism"

The patients was a lung neoplastic smoker: the lung alteration can easily be related to this condition. Even RA seems to be related to the smoke habit. In order to understand if a short-standing RA could have contribute to the lung alteration it is mandatory to know the evolution of disease activity.

Thank you for your insightful feedback!

The patient had no history of lung neoplasia but was diagnosed with basal cell carcinoma, a type of skin cancer.

While we acknowledge that the patient’s smoking history could also contribute to lung alterations, smoking being associated with respiratory bronchiolitis, desquamative interstitial pneumonia-like (DIP-like) changes, smoking-related interstitial fibrosis (SRIF) and emphyzema. Vascular remodeling may be observed in smokers, even in the absence of chronic obstructive pulmonary disease (COPD). SRIF is a distinct form of hyalinized collagen deposition in the interstitial compartment, predominantly affecting subpleural and centrilobular regions, with minimal clinical symptoms. This differs from our patient’s case, which involved extensive fibrosis with significant clinical manifestations.

Regarding rheumatoid arthritis (RA), it is indeed recognized that smoking is a major environmental risk factor for RA development and can exacerbate its progression. The multifactorial nature of the patient's condition made very difficult to determin the extent to which each factor could have contributed to these lung alterations. More  information on the disease's activity, such as inflammatory markers or autoantibody profiles were unavaible for us, as the rheumatological examination of the patient was conducted in another clinic.

Thank you!

Reviewer 2 Report

Comments and Suggestions for Authors

Dear Editor,

The authors aim to present Multiple Pulmonary Involvement in the Rapidly Progressive Evolution of Rheumatoid Arthritis. Although it contains valuable images in terms of educational material, it does not contribute recent data to the literature.

Best Regards

Author Response

Reviewer 2

Dear Editor,

The authors aim to present Multiple Pulmonary Involvement in the Rapidly Progressive Evolution of Rheumatoid Arthritis. Although it contains valuable images in terms of educational material, it does not contribute recent data to the literature.

Best Regards

Thank you for your insightful feedback!

Through the imaging analysis, corroborated with the patological interpretation , this case brings important directions for further research, because patients cannot be "cased" in rigid diagnostic algorithms, only because medical research has failed to determine statistically sufficient variables for rare cases. The association of the 2 aspects of UIP and NSIP imposes on the medical world an open approach, which allows the diagnosis of overlap syndromes in interstitial pathology as well.

Thank you!

Reviewer 3 Report

Comments and Suggestions for Authors

The revised version of the manuscript still requires improvement. Chest CT demonstrated by the authors indicates lung fibrosis with reticular opacities, traction bronchiectasis and partly with honeycombing. They are dispersed both in cranio-caudal and in axial view. Thus, it is not true to describe them as peripheral and basal. In addition – extensive ground glass opacities are visible in the whole lungs. Such CT pattern illustrates end stage lung fibrosis with signs of acute worsening which may correspond to immunological worsening, infection or circulatory insufficiency.  This is indistinguishable from fibrotic AIP.

Based on the figures included, it is not possible to diagnose UIP or NSIP.

Chest CT in NSIP shows lung fibrosis localized in lower parts of the lungs, with a frequent sparing of sub-pleural region and also, with coexisting ground glass opacities. Honeycombing is not found in NSIP, it is the characteristic feature of  UIP. Chest CT in UIP patients should not include excessive ground glass opacities, but  they may be present at the time of acute disease worsening.  The authors may find the description of various CTD-ILD signs in recently published article -  Semin Respir Crit Care Med 2024;45:287304.

The histopathologic report should contain the conclusive remarks of the pathologist.

What type of lung disease is recognized by the pathologist?

             ILD due to methotrexate therapy is rare and is mostly described as NSIP.

              I feel that the article should be co-authored and revised by radiologist and pathologist.  

Author Response

Reviewer 3

The revised version of the manuscript still requires improvement.

Thank you for your insightful feedback!

I adjusted the description of the images with the expertise of a radiologist.

The histopathologic report should contain the conclusive remarks of the pathologist.

Pathology report was signed pulmonary thromboembolism and interstitial lung disease (UIP pattern and NSIP pattern) in the context of a patient with rheumatoid arthritis undergoing treatment.

What type of lung disease is recognized by the pathologist?

ILD due to methotrexate therapy is rare and is mostly described as NSIP.

The microscopic lesions exhibited extensive fibrosis and moderate lymphocytic inflammatory infiltrate within the interstitium, traction bronchiectasis – NSIP pattern, peribronchiectatic pneumonia, along with enlarged respiratory spaces, some of which were bordered by significant fibrotic tissue (‘honeycombing’) – UIP pattern.

We mentioned in the article the potential methotrexate therapy implications.     

Thank you!

Round 3

Reviewer 1 Report

Comments and Suggestions for Authors

- Please make the reader aware that the "perspective" of a MTX role in RA-ILD onset is less supported than the other one. In https://pubmed.ncbi.nlm.nih.gov/32646919/, the conclusion is "Our results suggest that MTX use is not associated with an increased risk of RA-ILD in patients with RA, and that ILD was detected later in MTX-treated patients". Moreover, consider these metanalysis (that are far more informative than case reports or (quite old) not systematic review):

https://pubmed.ncbi.nlm.nih.gov/37434169/, https://pubmed.ncbi.nlm.nih.gov/37434169/

- Add the lack of disease activity as a major bias that should be considered in every similar cases. 

Author Response

Please make the reader aware that the "perspective" of a MTX role in RA-ILD onset is less supported than the other one. In https://pubmed.ncbi.nlm.nih.gov/32646919/, the conclusion is "Our results suggest that MTX use is not associated with an increased risk of RA-ILD in patients with RA, and that ILD was detected later in MTX-treated patients". Moreover, consider these metanalysis (that are far more informative than case reports or (quite old) not systematic review):

https://pubmed.ncbi.nlm.nih.gov/37434169/, https://pubmed.ncbi.nlm.nih.gov/37434169/

  • Add the lack of disease activity as a major bias that should be considered in every similar cases.

R

Dear reviewer,

We make the change according to your request, it is marked in green.

Additionally, I would greatly appreciate you effort, all your observation it was relevant and useful.

Thank you

Reviewer 3 Report

Comments and Suggestions for Authors

Thank you for addressing all my remarks, the revised version of the manuscript deserves publication in Diagnostics.

Author Response

Thank you for addressing all my remarks, the revised version of the manuscript deserves publication in Diagnostics.

R-

I would greatly appreciate you effort, all your observation it was relevant and useful.

Thank you

Round 4

Reviewer 1 Report

Comments and Suggestions for Authors

All comments were addressed